# The Effects of Intra-Articular Triamcinolone and Autologous Protein Solution on Metabolic Parameters in Horses

**DOI:** 10.3390/ani14152250

**Published:** 2024-08-02

**Authors:** Allen E. Page, Mackenzie Johnson, Jordan L. Parker, Olivia Jacob, Rachel Poston, Amanda A. Adams, Emma N. Adam

**Affiliations:** Maxwell H. Gluck Equine Research Center, Department of Veterinary Science, University of Kentucky, Lexington, KY 40546, USAamanda.adams@uky.edu (A.A.A.); emma.adam@uky.edu (E.N.A.)

**Keywords:** triamcinolone, oral sugar test, thyrotropin-releasing hormone, endocrine, APS, metacarpophalangeal joint

## Abstract

**Simple Summary:**

Intra-articular corticosteroids are a popular treatment choice for joint-associated pain and inflammation in horses despite recent work on the metabolic effects of these drugs. The goal of this project was to compare metabolic effects between intra-articular triamcinolone acetonide and an autologous protein solution. Intra-articular triamcinolone acetonide caused significant changes in thyrotropin-releasing hormone test and oral sugar test results, which are commonly used tests for pituitary pars intermedia dysfunction and insulin dysregulation, respectively. The results suggest a period of at least 2 days and up to 7 days should elapse between a single 9 mg intra-articular triamcinolone acetonide treatment and these tests. Further, this controlled study found that intra-articular triamcinolone acetonide exhibits significant effects on adrenocorticotropin-releasing hormone, cortisol, glucose, and insulin, while the autologous protein solution does not. Importantly, the changes with intra-articular triamcinolone acetonide included significant increases in resting insulin, which could predispose horses to laminitis.

**Abstract:**

Intra-articular corticosteroids are a popular treatment choice for joint-associated pain and inflammation in horses despite recent work on the metabolic effects of these drugs. The goal of this project was to compare metabolic effects between intra-articular (IA) triamcinolone acetonide (TA) and an autologous protein solution (APS). Five mixed-breed geldings (4–9 years) were utilized for this project. Three identical and consecutive 28-day treatment blocks were used, with metacarpophalangeal IA treatments consisting of equal volumes of saline, a commercially available APS, or 9 mg of TA. Regular plasma and serum samples were collected for ACTH, cortisol, glucose, insulin, and thyroid hormone analysis, in addition to thyrotropin-releasing hormone (TRH) and oral sugar tests (OSTs). Significant treatment effects of IA TA were present at 48 h post-injection in both the TRH and the OST. There was also significant suppression by IA TA of baseline ACTH and cortisol between 2 h and 96 h post-treatment, hyperglycemia between 12 h and 48 h, and hyperinsulinemia at 32 h post-treatment. There were no treatment effects with respect to any measured thyroid hormones, nor were there any significant treatment effects of APS noted. Results suggest at least 2 days and up to 7 days should elapse between a single 9 mg IA TA treatment and OST and/or TRH testing. This study found that TA exhibits significant effects on ACTH, cortisol, glucose, and insulin, while the APS does not.

## 1. Introduction

The use of intra-articular medications in horses has been a mainstay of treating joint disease for decades. Corticosteroids are commonly used due to their low cost and ready availability, in spite of risks that include, but are not limited to, hypophyseal–pituitary–adrenal axis suppression and laminitis. While equine practitioners believe the most commonly used corticosteroid, triamcinolone acetonide, may be associated with laminitis [1], current research suggests that the risk for laminitis is relatively low in metabolically normal horses [2,3,4]. However, insulin dysregulation (ID) and pituitary pars intermedia dysfunction (PPID) put horses at increased risk for laminitis [5], and concerns remain regarding the administration of intra-articular corticosteroids to this group [6]. Importantly, PPID and ID are prevalent conditions in the equid [7,8,9,10], reinforcing the need to ensure that treatments for common joint conditions, such as osteoarthritis, are appropriate for this population.

In recent years, the use of intra-articular orthobiologic regenerative therapies has been adopted by the equine industry as an effective, non-steroid treatment option for joint pain and inflammation [1]. Further, recent work has demonstrated a significant effect of intra-articular triamcinolone acetonide on resting levels of insulin, glucose, ACTH, and cortisol in horses [6,11,12]. As such, these new orthobiologic modalities may provide a safer alternative to intra-articular corticosteroid administration. One such modality, autologous protein solution, is a commercially available product that has been evaluated as an intra-articular therapeutic for equine osteoarthritis [13]. Therefore, the goal of this project was to compare the metabolic effects of intra-articular triamcinolone acetonide with those of a commercially available autologous protein solution to better understand the potential side effects of these treatments in horses. Based on our prior work [11], the hypothesis for this project was that 9 mg of intra-articular triamcinolone acetonide would elicit changes in a variety of resting metabolic parameters, including insulin, glucose, ACTH, and cortisol, while the autologous protein solution would not. Further, we hypothesize that intra-articular triamcinolone would cause changes in the common dynamic tests for PPID and ID, the thyrotropin-releasing hormone test, and the oral sugar test, respectively.

## 2. Materials and Methods

Five (one 9yo, one 7yo, and three 4yo) mixed-breed geldings from the University of Kentucky’s Department of Veterinary Science research herd were utilized for this project, which was conducted between March and May. Information regarding the horses used in this study is provided in Table 1. Three weeks prior to the start of the study, the horses were screened using an oral sugar test (OST) and thyrotropin-releasing hormone (TRH) test (described below) to ensure they were metabolically normal. Horses were in normal body condition (body condition score = 5–6), free from common clinical signs associated with ID or PPID (abnormal adipose accumulation), clinically sound, and free from signs of infectious disease immediately prior to the start of the study. This work was approved by the University of Kentucky’s Institutional Animal Care and Use Committee (#2023-4199).

Throughout the study, all 5 horses were co-housed on grass pasture with ad lib access to grass hay. In the 90 days prior to the start of the study, no horses were in exercise, nor had they received any anti-inflammatory treatment. Horses were provided approximately 1 kg of a balanced, pelleted, complete feed once a day to allow for visual inspection while maintained on pasture. The afternoon preceding intra-articular injections, horses were individually stalled inside, fed the same ration of hay, and subsequently returned to their group pasture after collection of the 24 h post-injection timepoint sample.

This study was divided into three identical and consecutive 28-day blocks, during which horses were injected, sampled, and completed a 14-day stand-down period, with previous work demonstrating normalization of metabolic parameters by 14 days after intra-articular triamcinolone acetonide [11]. Designed as a three-way crossover study, treatments were randomly assigned for Treatment Blocks #1 and #2 using a random number generator (Microsoft Excel, Office 365), with the final treatment being the remaining treatment for the horse. The injection leg was randomly assigned for Treatment Block #1 and then alternated for the remaining blocks.

Treatments consisted of an aseptic injection into one of the front metacarpophalangeal joints of 3 mL of 0.9% sodium chloride (negative control), 3 mL of an autologous protein solution (APS)(Pro-Stride APS®, Zoetis, Parsippany, NJ, USA), or 9 mg of triamcinolone acetonide (TA) with 2.1 mL of 0.9% sodium chloride (3 mL total volume). Horses assigned to the APS group had 60 mL of jugular venous blood collected aseptically approximately 30 min prior to joint injection, which was processed using the commercially available APS purification system per the manufacturer’s recommendations. Immediately prior to the joint injection, horses were sedated with xylazine (approximately 0.25 mg/kg IV).

Current guidelines from the Equine Endocrinology Group [14,15] were utilized for the OST and TRH testing. Briefly, horses were weighed the day prior to each treatment block, and all blood sample collections were performed using direct jugular venipuncture. TRH testing was always conducted first, and both dynamic tests were completed before any pelleted feed was provided on that day. TRH sample collections consisted of an initial (PRE) blood sample for adrenocorticotropin-releasing hormone (ACTH) analysis, followed by intravenous administration of 1 mg of TRH (Protirelin (as Acetate) (TRH-Thyrotropin-Releasing Hormone) in AQ vehicle, Wedgewood Pharmacy, Swedesboro, NJ USA), and a second blood sample collected exactly 10 min after the TRH administration (POST). The same lot of TRH was used throughout the entire study. To avoid unnecessary venipuncture procedures, the POST sample for TRH testing also served as the PRE sample for the OST. Immediately after the PRE OST sample was collected for insulin analysis, horses were orally administered 0.15 mL/kg of lite corn syrup (Karo Syrup, Karo Foodservice, Oakbrook Terrace, IL, USA), and a second sample was collected 60 min post-administration (POST). Both tests (OST and TRH) were always conducted together and took place the day prior to IA treatment (Day-1) and at 48 h, 168 h, and 336 h post-IA injection for each treatment block (Figure 1).

In addition to OST and TRH testing, resting blood samples for ACTH, insulin, glucose, cortisol, T3, T4, and free T4 (by dialysis—FT4d) were collected immediately prior to intra-articular injection (PRE), as well as at 2, 4, 8, 12, 24, 32, 48, 72, 96, 120, 144, 168, and 336 h post-injection (Figure 1). These samples were collected prior to being fed pelleted feed (above) and while they had access to hay and/or grass pasture. Any samples collected for ACTH, insulin, or glucose (daily, OST, or TRH) were collected into EDTA tubes and immediately placed on ice. Once glucose was tested using an equine-validated hand-held glucometer (AlfaTrack 3, Zoetis, Parsippany, NJ, USA) [16], EDTA blood tubes were centrifuged at 1000× *g* for 10 min, after which plasma was removed and immediately frozen at −20 °C. Samples for cortisol, T3, T4, and FT4d testing were collected into non-treated blood collection tubes and allowed to clot for 30+ minutes before centrifugation at 1000× *g* 10 min, followed by serum removal and storage at −20 °C. Once all treatment blocks were completed, samples were sent on dry ice to the Cornell University Endocrinology Research Testing Service for blinded analysis of ACTH, insulin, cortisol, T3, T4, and FT4d.

Sample size was determined with a free online website [17] using data from previous work with intra-articular triamcinolone [11] and associated differences in serum cortisol concentrations between saline and triamcinolone treatments at 4 h post-injection. With α = 0.05, power = 0.80, and a mean difference of 2.908, a sample size of three horses was determined to be required. Due to concerns regarding unknown variables for the other metabolic parameters of interest, two additional horses were included in this study, which resulted in a calculated power of approximately 0.95 for serum cortisol.

SigmaPlot 15 (Systat Software, Inc., Palo Alto, CA) was used for all statistical analysis, including post hoc. Two-way repeated measures analysis of variance (ANOVA) (Holm-Sidak) was used to examine differences in analyte concentrations by timepoint, treatment group, and timepoint by treatment group interactions. Significant treatment effects of APS or TA were deemed to be present only if there was a significant difference between the control and the corresponding treatment timepoints, as well as between the pre-treatment timepoint and a specific timepoint within an individual treatment group. Results were considered significant if *p* < 0.05.

## 3. Results

### 3.1. Dynamic Testing

Treatment effects of intra-articular TA were present at 48 h post-injection in both the thyrotropin-releasing hormone and the oral sugar tests. These effects manifest as a decrease in both PRE and POST ACTH concentrations in the TRH test (Figure 2a) and an increase in POST insulin in the OST (Figure 2b). Intra-articular treatment with APS did not result in any significant differences for either test at any timepoint.

### 3.2. Resting Concentrations

There was significant suppression of resting ACTH and serum cortisol concentrations following IA TA between 2 h and 24 h, as well as 48 h to 96 h (Figure 3a,b, respectively). Resting glucose increased significantly following TA treatment between 12 h and 48 h (Figure 3c), while insulin was increased at 32 h, 72 h, and 120 h post-IA TA (Figure 3d). There were no treatment effects noted with respect to T3, T4, or FT4d concentrations (Figure 4a–c). Additionally, there were no significant treatment effects of APS noted for any measured analyte.

## 4. Discussion

Intra-articular corticosteroids are a popular treatment choice for joint-associated pain and inflammation in horses. While recent work by our group and others has started to explore the metabolic effects of these drugs, herein, we have provided a direct comparison of the metabolic effects between intra-articular triamcinolone acetonide and a commercially available autologous protein solution, including the effects of these different treatments on commonly used diagnostic tests for insulin dysregulation and pituitary pars intermedia dysfunction in horses.

A single 9 mg dose of TA in the metacarpophalangeal joint had a demonstrable effect on both OST and TRH tests conducted 48 h following treatment. While the noted effects on the TRH test likely arise from baseline suppression of ACTH, as seen starting at 2 h post-treatment and lasting through 96 h post-treatment, the expected post-TRH increase in ACTH at 48 h was significantly decreased. Additionally, the post-OST insulin response at 48 h was significantly higher in the TA-treated group. Importantly, average post-OST insulin concentrations were greater than 70 μIU/mL at 48 h following treatment with TA, a result well beyond the > 45 μIU/mL cutoff for the diagnosis of insulin dysregulation in horses [15]. This is particularly striking considering that this study utilized metabolically “normal” horses, and we are unaware of any prior work detailing the effects of an intra-articular corticosteroid on TRH or OST testing in horses. Based on these results, it is recommended that at least 2 days and up to 7 days be allowed to pass between IA TA administration and dynamic diagnostic testing for ID or PPID in horses. However, it stands to reason that this period may be different for higher doses of IA TA or other corticosteroids. Alternatively, given that the specific APS system utilized in this project did not alter any of the metabolic parameters that were evaluated, this APS system should be considered when dynamic diagnostic OST/TRH testing must be performed in close proximity to intra-articular treatment.

Beyond the effects of TA on OST and TRH testing, there were significant treatment effects of IA TA on resting levels of ACTH, cortisol, glucose, and insulin. Both ACTH and cortisol concentrations were significantly decreased between 2 h and 96 h post-TA treatment; the visible decrease in both ACTH and cortisol at 32 h was not considered a significant treatment effect due to the concomitant decrease in the control and APS groups, which is likely attributable to diurnal variations that are typically observed. Prior work in humans has demonstrated a substantial and prolonged suppression of ACTH concentrations following a single dose of triamcinolone [18], and similar cortisol suppression following IA treatment has been detailed in horses [11,12,19,20]. As noted previously [11], while side effects of cortisol suppression are not readily recognized in horses, prolonged decreases in serum cortisol concentrations following IA TA may induce immunosuppression and predispose horses to infection or leave them unable to respond to stressful events.

Concerns regarding the administration of corticosteroids and its association with the induction of laminitis are well reported in the literature and a concern to every equine veterinarian. While that risk is considered very low in metabolically normal horses [2,3,4], the risk is likely greater in equids with insulin dysregulation and/or PPID. As such, one of the goals of this study was to monitor the glucose and insulin response to IA treatment. Boger et al. [6] recently reported on the effects of 18 mg of IA TA in the middle carpal joint with regard to serum glucose and insulin concentrations [6]. In that study, they found that serum glucose significantly increased above baseline between 6 h and 48 h post-IA TA, while serum insulin was significantly increased above baseline at 6 h, 12 h, and 24 h. In contrast to the data presented here, they did not document any of their horses reaching resting insulin levels above 45 μIU/mL [15]. Hallowell et al. [12] recently detailed significant hyperinsulinemia and hyperglycemia in horses following injection with 9 mg of TA (18 mg total) into each radiocarpal joint [12]. Additionally, they reported that hyperinsulinemia was exacerbated in those horses with higher pre-treatment insulin concentrations, although a direct comparison of insulin results is difficult due to the use of a different testing method that was employed here and in Boger et al. [6]. In the study presented herein, increased glucose was present between 12 h and 48 h post-IA TA, and the average resting insulin was 71 μIU/mL at 32 h in the TA group. Further, resting insulin was >45 μIU/mL between 72 h and 120 h in the TA group, while neither of the other treatment groups exceeded insulin > 45 μIU/mL. The reasons for the different results between this study and Boger et al. [6] are not readily apparent, especially given the lower dose of TA used in this study, but potential explanations include different management/housing, seasonal variations in insulin sensitivity [21], different joint injected, and this study taking part on pasture during the spring growing season vs. horses being stalled throughout the Boger et al. [6] study. Nevertheless, in contrast to those presented by Boger et al. [6] and in agreement with Hallowell et al. [12], the results presented here demonstrate that the intra-articular administration of TA can induce a period of significant hyperinsulinemia, and concerns regarding the relationship between intra-articular corticosteroid-induced hyperinsulinemia and laminitis remain well justified. The increasing availability of point-of-care testing for hyperinsulinemia provides a significant and important tool for equine practitioners to consider prior to administering IA corticosteroids. Further, we echo Boger et al. [6] that expanding this work into insulin-dysregulated horses is the logical next step to better understanding the effect of these common treatments in an increasingly prevalent segment of the horse population. Critically, these results suggest that the APS product tested here may be a safer treatment option for horses with metabolic derangements, such as ID or PPID, although confirmatory testing in these specific populations is warranted. This is particularly important given the reported effects that exercise, stress, and medications may have on orthobiologic yields [22,23,24].

In an attempt to further examine the metabolic effects of intra-articular treatments, thyroid hormone analytes (T3, T4, and FT4d) were also quantified. While there were occasional timepoints in which resting levels of one or more analytes varied significantly from their PRE treatment value, there were no statistically significant changes directly attributable to treatment. This stands in contrast to previous work with topical dexamethasone administration, in which significant suppression of T3 and T4 was noted [25]. Possible explanations may be that testing parameters between these studies differed and, in the other study, reverse T3 and free T3 were measured, both of which have previously been shown to increase following intra-muscular dexamethasone administration in horses [26].

Several limitations of this study were identified, including that the investigators were not blinded to the treatment groups. However, as a study that depended solely on quantitative outcomes, this did not affect the results presented here. Following APS purification, the produced solution was not subjected to measurement of any cytokines or growth factors. As a result, individual horse variability could have led to varying amounts of the concentrated proteins. Current guidelines for OST sampling support the collection of a 60 min and/or 90 min post-sample. This study utilized only a 60 min post-sample, and there is a possibility that a 90 min sample or the use of the higher 0.45 mL/kg dose of oral sugar may have provided a different result than that reported here. An additional potential limitation was the use of only 9 mg of triamcinolone acetonide, although this dose was based on our previous work [11] and is well represented in the literature [27,28]. Given that the concurrent treatment of multiple joints and/or the use of higher doses are common in equine practice [1], it might be expected that the treatment effects reported here would be more pronounced had a larger cumulative dose of TA been used in this study.

## 5. Conclusions

This study found intra-articular triamcinolone acetonide-induced changes in commonly used, dynamic diagnostic tests for insulin dysregulation and pituitary pars intermedia dysfunction in horses. The data suggests that a period of up to 7 days should elapse between intra-articular TA treatment and OST and/or TRH testing, based on a single IA dose of 9 mg. Further, this controlled study presented a comparison of two common intra-articular treatments in horses and demonstrated that triamcinolone acetonide exhibits significant metabolic side effects, while the commercially available autologous protein solution used here does not.

## Figures and Tables

**Figure 1 animals-14-02250-f001:**
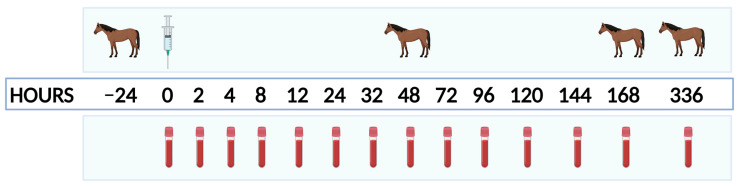
Timeline representation of each treatment block. Horse icons represent timepoints at which thyrotropin-releasing hormone and oral sugar testing were carried out. The syringe icon represents the timepoint at which that block’s intra-articular treatment was administered. The blood tube icons represent when whole blood was collected for resting concentrations of ACTH, insulin, glucose, cortisol, T3, T4, and free T4 (by dialysis—FT4d).

**Figure 2 animals-14-02250-f002:**
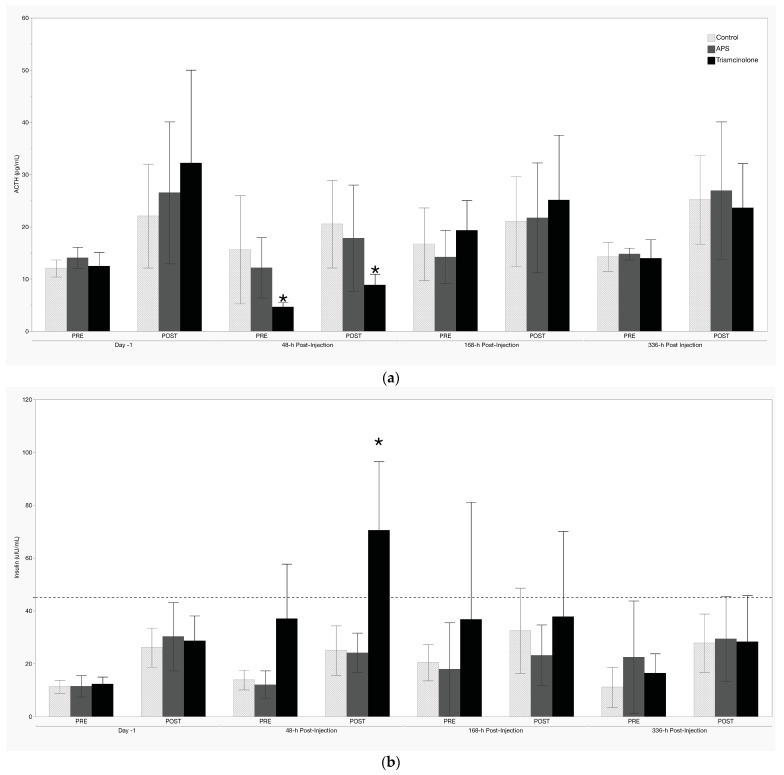
(**a**) Thyrotropin-releasing hormone and (**b**) oral sugar testing results. Values shown are mean +/− standard deviation. The dashed line in (**b**) represents the currently recommended cutoff for the diagnosis of ID [15]. * denotes a significant treatment effect (*p* < 0.05).

**Figure 3 animals-14-02250-f003:**
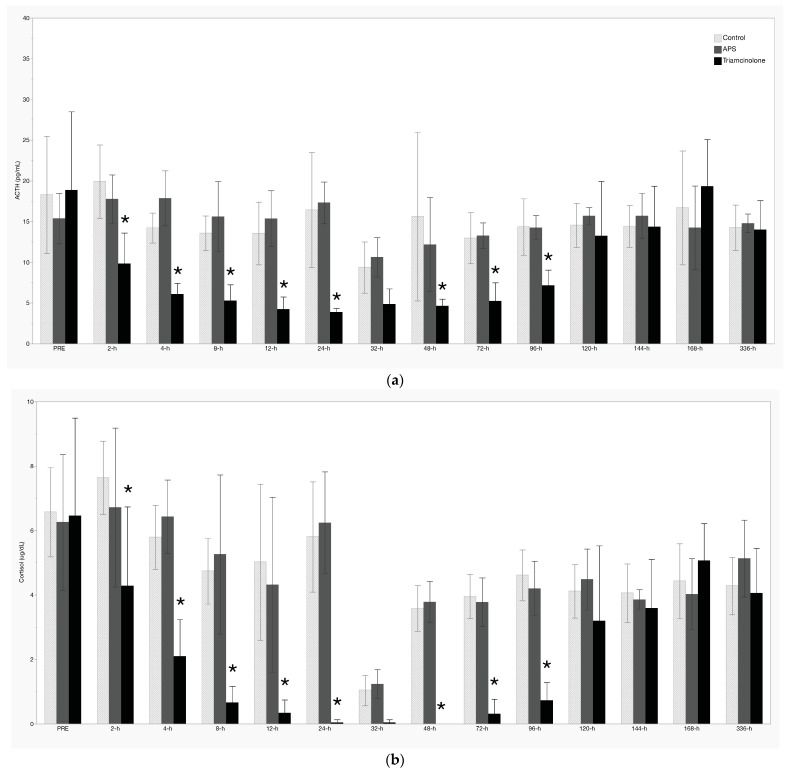
Resting concentrations of (**a**) adrenocorticotropin-releasing hormone (ACTH), (**b**) cortisol, (**c**) glucose, and (**d**) insulin. Values shown are mean +/− standard deviation. * denotes a significant treatment effect (*p* < 0.05).

**Figure 4 animals-14-02250-f004:**
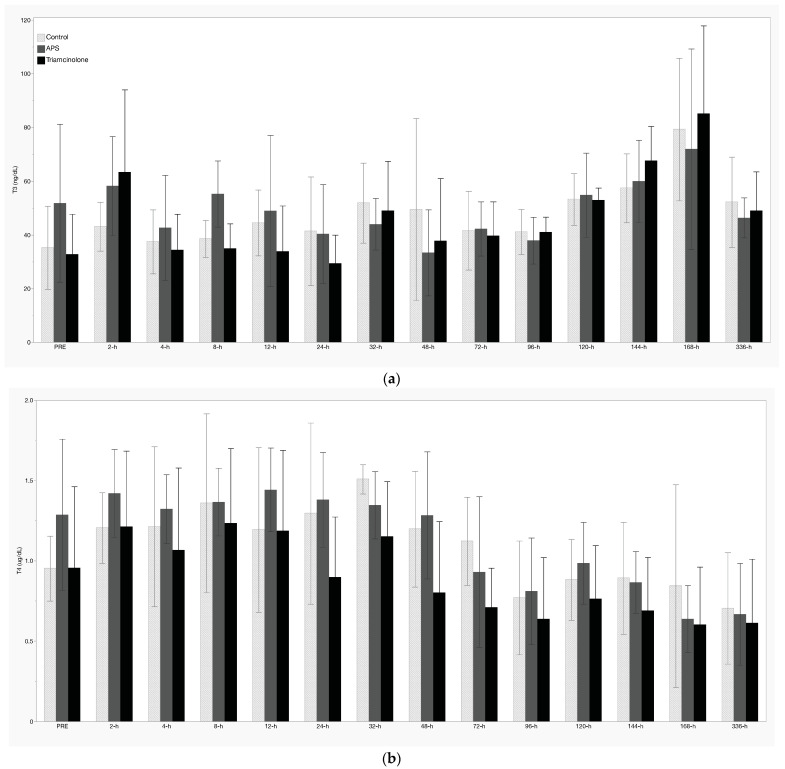
Resting concentrations of (**a**) T3, (**b**) T4, and (**c**) free T4 (by dialysis). Values shown are mean +/− standard deviation. No significant treatment effects were noted (*p* < 0.05).

**Table 1 animals-14-02250-t001:** Demographic information and baseline insulin concentration.

	Mean ± Standard Deviation
Age (years)	5.6 ± 2.3
Body Condition Score (0–9)	5.5 ± 0.5
Weight (kg)	595 ± 55.1
Insulin Concentration (Pre-Treatment #1—μIU/mL)	14.2 ± 2.35

## Data Availability

Data utilized in this manuscript are available at https://dx.doi.org/10.6084/m9.figshare.26190467.

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
