# Peer review of "The Effects of Intra-Articular Triamcinolone and Autologous Protein Solution on Metabolic Parameters in Horses"

_animals, 2024, doi:10.3390/ani14152250_

Round 1
Reviewer 1 Report
Comments and Suggestions for Authors
Overall this is a well designed and well written study but under-referenced and discussed in regards to APS which is the more novel part of the study since several recent TA papers on metabolic effects.
There needs to be more of a discussion on the inherent variability of autologous APS vs TA and also the fact that APS can be affected by medications the horse is on, exercise, etc. Were any of the horses used in this study on medications in recent past? As reads currently sounds like APS is all the same thing which it is not.
Existing studies in literature on variability of APS, possible effects of NSAIDs, and exercise etc (some published on ACS but very similar) should at least be mentioned. Also should be a limitation that APS not characterized meaning how do you know that you actually made APS from each horse if didnt measure cytokines etc. Some runs do fail and again there is a lot of variability between horses.
Entire discussion really focused on TA and neglects APS. Once again, conclusion makes it sound like commercially available APS is a single entity that is always the same - using 5 healthy horses free of medications to make APS is likely not representative of clinical population on medications, supplements, etc and this should at least be acknowledged.
Several of the figures are missing legends - please check these.
Reviewer 2 Report
Comments and Suggestions for Authors It's a good article with the limitations well described at the end, I would just like to know if the sample power test was carried out to know if the number of animals was enough to obtain reliable results. It would be good to include this information in the methodology.
Reviewer 3 Report
Comments and Suggestions for Authors
Congratulations for the well designed work.
Some few comments: the use of the term "block" to describe the treatment groups can lead to some misinterpretation. The APS product was subjected to any kind of evaluation? Specially regarded to Insulin Like Growth Factor (IGF). Even that none measurement could be done on the orthobiologic used, its important to mention the presence of its average molecular content.
Comments on the Quality of English Language
Good quality of writing. Maybe the authors could standardize some measures used, in some parts of the manuscript hours is presented in different forms, as 2-h and 98-hr.
